# Galectin-1 and Galectin-3 in B-Cell Precursor Acute Lymphoblastic Leukemia

**DOI:** 10.3390/ijms232214359

**Published:** 2022-11-18

**Authors:** Fei Fei, Mingfeng Zhang, Somayeh S. Tarighat, Eun Ji Joo, Lu Yang, Nora Heisterkamp

**Affiliations:** 1Section of Molecular Carcinogenesis, Department of Pediatrics, Division of Hematology/Oncology and Bone Marrow Transplantation, The Saban Research Institute of Children’s Hospital, Los Angeles, CA 90027, USA; 2Department of Systems Biology, Beckman Research Institute City of Hope, Monrovia, CA 91016, USA

**Keywords:** acute lymphoblastic, Bcr/Abl, combination drug treatment, double knockout, Galectin-1, Galectin-3, GM-CT-01, GR-MD-02, microenvironment, OP9 stromal, sialic acid, ST6Gal1, migration, MYH9

## Abstract

Acute lymphoblastic leukemias arising from the malignant transformation of B-cell precursors (BCP-ALLs) are protected against chemotherapy by both intrinsic factors as well as by interactions with bone marrow stromal cells. Galectin-1 and Galectin-3 are lectins with overlapping specificity for binding polyLacNAc glycans. Both are expressed by bone marrow stromal cells and by hematopoietic cells but show different patterns of expression, with Galectin-3 dynamically regulated by extrinsic factors such as chemotherapy. In a comparison of Galectin-1 × Galectin-3 double null mutant to wild-type murine BCP-ALL cells, we found reduced migration, inhibition of proliferation, and increased sensitivity to drug treatment in the double knockout cells. Plant-derived carbohydrates GM-CT-01 and GR-MD-02 were used to inhibit extracellular Galectin-1/-3 binding to BCP-ALL cells in co-culture with stromal cells. Treatment with these compounds attenuated migration of the BCP-ALL cells to stromal cells and sensitized human BCP-ALL cells to vincristine and the targeted tyrosine kinase inhibitor nilotinib. Because N-glycan sialylation catalyzed by the enzyme ST6Gal1 can regulate Galectin cell-surface binding, we also compared the ability of BCP-ALL wild-type and ST6Gal1 knockdown cells to resist vincristine treatment when they were co-cultured with Galectin-1 or Galectin-3 knockout stromal cells. Consistent with previous results, stromal Galectin-3 was important for maintaining BCP-ALL fitness during chemotherapy exposure. In contrast, stromal Galectin-1 did not significantly contribute to drug resistance, and there was no clear effect of ST6Gal1-catalysed N-glycan sialylation. Taken together, our results indicate a complicated joint contribution of Galectin-1 and Galectin-3 to BCP-ALL survival, with different roles for endogenous and stromal produced Galectins. These data indicate it will be important to efficiently block both extracellular and intracellular Galectin-1 and Galectin-3 with the goal of reducing BCP-ALL persistence in the protective bone marrow niche during chemotherapy.

## 1. Introduction

All cells are covered by a dense layer of glycans attached to proteins and lipids. The glycans form docking sites for carbohydrate-binding proteins called lectins [1]. Galectin-1 and Galectin-3 are members of a family of 9 lectins that specifically recognize Gal-GlcNAc (LacNAc) branches of N-glycans attached to cell surface glycoproteins [2]. Binding of Galectin-1 and Galectin-3 to glycoprotein clients can be inhibited by the further capping of the LacNAc branches by the glycan sialic acid [2]. For N-linked glycans, this addition is catalyzed in the Golgi by one main sialyltransferase, ST6Gal1 [3]. Apart from their activity on the cell surface, Galectin-1 and Galectin-3 also are expressed intracellularly at different locations and have functions that may be independent of their carbohydrate-binding activity [4].

Galectin-1 and Galectin-3 overexpression has been widely associated with worse outcomes in many cancers [5], in which they can be expressed by both cancer cells as well as non-cancer stromal cells [6,7,8]. BCP-ALL constitutes a group of developmentally arrested, immature B-lineage precursors that can be categorized into numerous subgroups on the basis of genetic abnormalities [9,10,11,12]. Some BCP-ALLs are driven by mutated transcription factors, such as the fusion of Etv6 and Runx1 [13], or recombination of different genes with the MLL gene [14]. Ph-chromosome-positive ALL (Ph-positive ALL) is one major poor-prognosis subcategory of ALL, characterized by the presence of a t(9;22) translocation that fuses two genes, *BCR* and *ABL*, at the breakpoints [15]. The chimeric Bcr/Abl protein that is produced as a result has deregulated tyrosine kinase activity that can be inhibited by targeted small molecule inhibitors such as nilotinib [16].

All BCP-ALLs, regardless of driver mutations, develop at a common anatomical site, the microenvironment of the bone marrow. This is also the most frequent location of relapse during or after chemotherapy [17]. In fact, the development of drug resistance by cancer cells is actively supported by their microenvironment [18,19]. We and others [20,21,22,23,24,25,26,27,28] model BCP-ALL drug resistance development ex vivo by co-culture with OP9 bone marrow stromal cells. OP9 cells are a major source of SDF-1α, the main chemokine attracting leukemia cells and promoting their association with bone marrow stromal cells [29,30]. Rellick et al. [31] recently reported that leukemia cells in such co-cultures have a gene expression signature that mimics that of cells remaining in patients after induction therapy. The migration towards and adhesion to bone marrow stroma is one mechanism that contributes to chemoprotection provided by stromal cells to BCP-ALL [32,33], but overall, the molecular mechanisms through which such bone marrow stromal cells promote BCP-ALL drug resistance are incompletely understood.

There is increasing evidence that Galectin-1 and Galectin-3 are involved in the protective cross-communication between leukemia cells and the bone marrow microenvironment [34,35,36]. We previously determined that Galectin-1 and Galectin-3 each promote migration and adhesion of BCP-ALL cells to stromal cells [8,25] and showed that drug treatment of BCP-ALL cells also induces the production of endogenous Galectin-3 [27]. Juszczynski et al. previously reported that expression of Galectin-1 is restricted to a specific subclass of BCP-ALL cells characterized by MLL rearrangements [37]. We found that Galectin-1 is also present in other subcategories of BCP-ALL including Ph-positive ALL [28]. Thus, the Galectin-1 and Galectin-3 that could protect BCP-ALL cells during drug treatment is complicated in terms of origin, and both leukemia cells as well as stromal cells are a possible source of these Galectins.

Inhibition of Galectin-1 or Galectin-3 activity is likely to be beneficial as a method to chemosensitize BCP-ALL cells and reduce or eliminate residual leukemia cells. However, the possible overlap in Galectin-1 and Galectin-3 expression and function could be a confounding factor since the expression of one could compensate for the inhibition of the other. To gain insight into this issue, we here further investigated how Galectin-1 and Galectin-3 together promote survival of BCP-ALL cells. Our results support the concept that simultaneous blocking of Galectin-1 and Galectin-3 would be a preferred strategy to potentiate the activity of drugs used to treat BCP-ALL.

## 2. Results

### 2.1. Galectin-1 and Galectin-3 Are Concurrently Expressed in Many Normal and Abnormal Hematopoietic Cell Types as Well as Stroma

Expression of Galectin-1 and Galectin-3 in the same cells appears to be relatively common, although their levels and expression patterns differ in some cases. For example, normal human hematopoietic cells (Appendix A) express both Galectin-1 and Galectin-3. Murine B-cell precursor *Lgals1* appears to decrease during B-cell maturation (Appendix A), whereas *Lgals3* levels are relatively constant (Appendix A). Activated and in-vitro-expanded NK cells show an increase in *LGALS1* but not *LGALS3* expression, whereas only *LGALS1* is upregulated in AML cells compared to normal CD34+ HSC (Appendix A).

We also assessed their co-expression in Ph-positive BCP-ALL cells. The majority of samples in two gene array data sets [38,39], representing a total of 33 primary cases of Ph-positive ALL, simultaneously expressed both *LGALS3* and *LGALS1* mRNAs (not shown). Interestingly, GSE79533 showed a positive correlation (R^2^ = 0.46 and *p* < 0.01) between expression of these two genes, suggesting that in some cases their expression may be linked (Figure 1a). In the ETV6-RUNX1 subcategory of BCP-ALL, expression was also positively correlated (R^2^ = 0.26 and *p* < 0.0001, R^2^ = 0.16 and *p* < 0.01) (Figure 1b), but a similar analysis of MLL-rearranged BCP-ALLs showed no correlation, indicating that this may be subtype dependent.

Mouse models for Ph-positive ALL include transgenic mice that develop ALL within around 3 months after birth [40]. We analyzed an RNA microarray gene expression data set of such samples (GSE110104) to determine if *Lgals1* and *Lgals3* are endogenously expressed in the mouse leukemia cells. Samples included flow-sorted bone marrow cells from matched wild-type; preleukemic; fully leukemic; and fully leukemic mice treated for 8 days with nilotinib, a targeted Bcr/Abl kinase inhibitor [41,42]. Whereas the absolute values for *Lgals1* expression were higher than those for *Lgal3* (compare MFI values in the right and left panels in Figure 1c), both non-leukemic wild-type and leukemic B-cell precursor samples expressed both Galectins. Expression levels of *Lgals1* were comparable in all samples. Consistent with our previous results in human BCP-ALL cells treated with drugs [27], the expression of *Lgals3* was significantly induced in drug-treated BCP-ALL cells that had been isolated from mice treated for 8 days with 75 mg/kg nilotinib.

We also assessed Galectin-1 and Galectin-3 protein expression by FACS on primary normal human bone marrow mesenchymal stromal cells (MSCs). This analysis showed that such MSCs express high levels of both Galectins (Appendix A). Similarly, on the basis of analysis of GSE56472 [43], OP9 bone marrow stromal cells were found to contain both Galectin-1 and Galectin-3 (Appendix A, OP9-WT lane). However, on the basis of the pattern of gene expression in OP9 cells treated with imatinib (Appendix A), *Lgals1* transcription was stable, whereas that of *Lgals3* appeared to be dynamically regulated. We conclude that although Galectin-1 and Galectin-3 are expressed in both hematopoietic and stromal cells, regulation of their expression is distinct.

### 2.2. Inhibition of Exogenous Galectins Inhibits Migration

Because stromal cells can be a significant source of these Galectins to BCP-ALL cells, we sought to determine the effect of inhibition of extracellular Galectin-1 and Galectin-3. For this, we used GM-CT-01 and GR-MD-02, also known as Davanat and Belapectin, respectively. These relatively high-molecular-weight (50–60 kDa) carbohydrates derived from natural plant glycans [44,45,46,47] presumably do not enter cells, as reviewed in [48]. Since the in vitro K_d_ of both compounds (GR-MD-02: Galectin-1 10 μM, Galectin-3 2.9 μM; GR-CT-01: 8 μM and 2.8 μM, respectively) was reported to be comparable [48], they could inhibit both extracellular Galectin-1 and Galectin-3.

We first tested GM-CT-01 on K562, a chronic myeloid leukemia cell line that constitutively produces both endogenous Galectin-3 and Galectin-1 [25,49]. As shown in Figure 2a,b, when these cells were plated on mouse embryonic fibroblasts (MEFs) and the samples were run on a lower percentage SDS-PAA gel, two Galectin-3 bands were visible. The lower band is human Galectin-3, whereas that with a slightly larger molecular mass is murine Galectin-3 produced by the MEFs. The cells were treated with GM-CT-01 to determine if this would change levels of extracellular (mouse)/intracellular (human) Galectin-3. Interestingly, treatment with GM-CT-01 for 2 h reduced the signal from mouse Galectin-3 (top band, compare lanes 3 and 4 in Figure 2a,b), supporting the idea that this compound inhibits binding of extracellular Galectin-3 to leukemia cells. Treatment with GM-CT-01 for 24 h also decreased levels of Galectin-3 made by K562 cells, both when they were plated alone (Figure 2b, compare lanes 1 and 3) as well as when they were plated on MEFs. MEF-produced Galectin-3 was also reduced by GM-CT-01 treatment (upper band lanes 4 and 5).

To confirm that GM-CT-01 reduces binding of extracellular Galectin-3 to BCP-ALL cells, we harvested BCP-ALL cells from underneath the stroma and stained the leukemia cells for Galectin-3 with or without prior incubation with GM-CT-01. As shown in Figure 2c, binding of Galectin-3 antibody to the cells was reduced in the presence of GM-CT-01, suggesting that the compound is able to decrease exogenous cell-surface-bound Galectin-3 on these cells.

Galectin-1 is involved in promoting the survival of many cancers including hematopoietic malignancies [50,51,52,53,54]. BCP-ALL cells also express cell surface glycoproteins that are ligands for extracellular Galectin-1 (Appendix A). We previously showed that some Galectin-1 is found outside the cells in medium conditioned by BCP-ALL/OP9 co-cultures and in OP9-produced exosomes [27,28]. However, depending on the exact BCP-ALL and possibly co-culture conditions, the percentage of cells with Galectin-1 on the surface varies [28]. We treated US7 and TXL2 cells with GM-CT-01 or GR-MD-01 to examine reduction of possible cell-surface-associated Galectin-1 but found no clear reduction in total Galectin-1 (Appendix A).

We next investigated the effect of these compounds on BCP-ALL cell migration. Stromal cells produce SDF-1α, the main chemoattractant for BCP-ALL cells [24,55]. Therefore, when BCP-ALL cells are introduced into a Transwell that has a stromal layer on the bottom, they migrate into the bottom compartment. Figure 2d,e shows that at high concentrations, GM-CT-01 reduced migration of TXL2 and US7 towards OP9 stroma. GR-MD-02 (Figure 2f,g) was active in the inhibition of cell motility at lower concentrations than GM-CT-01.

### 2.3. Effect of GM-CT-01 or GR-MD-02 on Intracellular BCP-ALL Signal Transduction and Proliferation in the Presence of Stromal Support

We also analyzed if GM-CT-01 or GR-MD-02 treatment affects intracellular signaling. Figure 3a shows that a 3-day treatment with GM-CT-01 attenuated pErk1/2 signals in both US7 and TXL2 cells and also reduced endogenous levels of p-p38. GR-MD-02 (Figure 3b), similar to GM-CT-01, reduced the phosphorylation of p38 and Erk1/2, but Akt and STAT5 phosphorylation or cytoplasmic NF-kB p-p65 and NF-kB p65 levels were not clearly affected after long-term exposure to GR-MD-02 (Figure 3c).

We then measured whether GM-CT-01 alone or in combination with therapeutic drugs affects the viability of BCP-ALL cells in the presence of OP9 stromal cells. As shown in Figure 4a,b, whereas GM-CT-01 alone had no obvious effect on TXL2 / US7 viability or cell proliferation, the addition of GM-CT-01 to nilotinib/vincristine-treated cells further decreased cell viability and cell numbers beyond mono-treatment. As shown in Figure 4c, consistent with these results, US-7OP9 co-cultures treated with different concentrations of GM-CT-01, or with vincristine as mono-treatment, were morphologically similar to controls, showing a mix of phase-bright and phase-dim leukemia cells on a background of elongated adherent stromal cells. However, treatment with both vincristine and 5 or 10 mg/mL GM-CT-01 was associated with increased numbers of de-adhered, phase bright leukemia cells in the media (Figure 4c).

GR-MD-02 had an effect at lower concentrations than GM-CT-01. Similar to GM-CT-01, the compound did not reduce cell viability when used as monotreatment (Figure 4d,e, left panels GR-MD-02 alone) but did inhibit proliferation of both TXL2 (Figure 4d, right panel) and US7 (Figure 4e right panel) as measured by a reduction in the total viable cell count. Nilotinib as a tyrosine kinase inhibitor is not extremely cytotoxic, but its combination with relatively lower amounts of GR-MD-02 (Figure 4d, right panel, 1 mg/mL and lower) produced a significant reduction in cell counts. GR-MD-02 with vincristine (Figure 4e left panel) also had increased toxicity at 2.5 and 5 mg/mL compounds and a cytostatic effect on the US7 ALL cells at 5 mg/mL (Figure 4e, right panel). Thus, these results show that interference with Galectin binding to BCP-ALL cells sensitizes the cells to chemotherapy.

### 2.4. Galectin-1 × Galectin-3 Double Null Mutant BCP-ALL Cells Have Decreased Proliferation and Survival

To investigate the effect of simultaneous loss of endogenous Galectin-3 and Galectin-1 function on BCP-ALL cells, we transformed primary B-lineage murine bone marrow *Lgals1 × Lgals3 -/-* (hereafter referred to as dKO) and wild-type (wt) cells [56] with the p190 Bcr/Abl oncogene to generate an aggressive, well-proliferating B-cell precursor ALL [57] that does not depend on stromal support. Early passage immunophenotyping of two independently transduced sets of wt and dKO BCP-ALL cells showed they were similar in that they lacked surface IgM but were positive for CD43, B220, and AA4.1 (Figure 5a). Samples were heterogeneous for the expression of the BAFF-R and CD24. Western blotting of early passage cells grown without stroma confirmed the lack of Galectin-1 (Figure 5b).

As expected, on the basis of the expression of the Bcr/Abl tyrosine kinase, these cells contained many tyrosine phosphorylated proteins (Figure 5b, pY20 Western blot panel) as a consequence of the deregulated tyrosine kinase activity of this oncoprotein, but there were no clear differences between wt and dKO cells in commonly activated signal transduction pathways in this type of leukemia cells including Stat5, Erk, p38, or Akt. However, levels of p-Src were increased in the dKO cells. As shown in Figure 5c, lack of endogenous Galectin-1 and Galectin-3 also reduced the migration of mouse Bcr/Abl-expressing BCP-ALL cells towards MEFs, indicating that cell-endogenous Galectins contribute to efficient migration.

Physiologically, wt and dKO BCP-ALL cells were clearly different. Whereas wild-type BCP-ALL cells had robust cell growth, loss of both endogenous Galectin-1 and -3 resulted in considerably reduced proliferation rates (Figure 6a). Using murine BCP-ALL *Lgals3-/-* cells, we previously showed that the ability to produce endogenous Galectin-3 provides protection to the BCP-ALL cells against chemotherapy [26]. Pharmacological inhibition studies also indicated that endogenous Galectin-1 provides chemoprotection [28]. Therefore, we compared the growth and survival of wt murine BCP-ALL cells to that of the dKO cells under drug treatment.

Cells were continuously treated with a non-lethal dose of vincristine, one component of the standard induction chemotherapy regimen for BCP-ALL. As shown in Figure 6b, wt2 or wt1 BCP-ALL cells treated with vincristine showed an initial drop in viability (panels (1) and (3)) but resumed proliferation (panels (2) and (4)) as the cells became resistant to the chemotherapy. However, the viability (panels (1) and (3)) of dKO2 or dKO1 BCP-ALL cells treated with vincristine did not recover after 4–5 days of drug exposure, and no cell growth (panels (2) and (4)) was measured.

Stromal cells (including MEFs) provide significant protection to human BCP-ALL cells against many types of drug treatment [20,22,23,24]. To examine if co-culture with stromal cells could compensate to any degree for lack of endogenous Galectin-1 and Galectin-3 in double null mutant cells, we compared the growth and survival of dKO BCP-ALL cells, cultured with and without MEFs during nilotinib treatment. As shown for wt1 (Figure 6c, panel (1)) and wt2 (Figure 6d, panel (1)), consistent with their stromal independence, wt BCP-ALL cells had similar viability with or without co-culture with MEFs. Moreover, viability was minimally affected by nilotinib treatment. Nilotinib did inhibit the proliferation of wt1 (Figure 6c, panel (2)) and wt2 (Figure 6d, panel (2)), but the presence of MEFs had no positive effect on proliferation. For dKO1 (Figure 6c, panel (3)) and dKO2 (Figure 6d, panel (3)), nilotinib treatment decreased cell viability, but the presence of MEFs significantly provided protection against the drug. Thus, some of the cells survived over the duration of the treatment. However, there was still little proliferation (Figure 6c,d panels (4)). Previously, we showed that loss of Galectin-3 alone in such cells has little effect on normal proliferation [26]. On the basis of the current data, it seems that Galectin-1 is able to compensate, to some degree, for lack of Galectin-3 expression in Galectin-3 knockout cells.

### 2.5. Contribution of Stromal-Produced Galectin-1 to BCP-ALL Homeostasis

On the basis of these results and our previously published data [25,26,27], we conclude that both extracellular Galectin-3 as well as endogenous Galectin-3 + Galectin-1 contribute in a significant way to BCP-ALL survival under chemotherapy stress. However, such compounds do not allow for a discrimination between stromal-produced secreted and leukemia-produced secreted Galectins. We therefore compared the effect of OP9 stromal knockout of Galectin-1 [58] (Appendix A) or Galectin-3 [25] on the BCP-ALL cells.

Data from other types of cancers support a model in which the ability of extracellular Galectin-3 and Galectin-1 to bind to cell surface glycoproteins is regulated by the exact composition of the glycan structures attached to client proteins. Both Galectins recognize LacNAc, which can be present as a single structure or as LacNAc repeats attached to N-glycans or O-glycans. Importantly, when N-linked glycan cores are terminated by α2-6 sialylation, Galectin-1 no longer binds to client glycoproteins (see schematic Appendix A), whereas Galectin-3 engagement depends on whether the α2-6 sialylated N-glycan structure contains one, or more than one, linear LacNAc monomers [2]. Seeing that ST6Gal1 is the sole enzyme outside the nervous system that can add sialic acid in such an α2-6 linkage to terminate N-linked glycan cores, we included Cas9/CRISPR-generated ST6Gal1 knockdown (KD) BCP-ALL cells in this analysis [58]. ST6Gal1 KD cells are predicted to have increased capacity for extracellular Galectin-1 (Appendix A) to bind to specific cell surface N-linked glycans. As shown in Appendix A, over a period of 15 days, the steady-state growth of two BCP-ALLs, JFK125 and JFK125R, was comparable, irrespective of the OP9 stromal support (Galectin-3 or Galectin-1 knockout) or the absence or presence of ST6Gal1 activity in the BCP-ALL cells. However, and consistent with our previous results [25], lack of Galectin-3 produced by OP9 cells significantly reduced the ability of JFK125 or JFK125R BCP-ALL cells to resist vincristine chemotherapy treatment (Figure 7a,b). Interestingly, a positive contribution of stromal Galectin-1 to the survival of these cells under continuous chemotherapy was only evident in the first days (JFK125, Figure 7c), whereas for JFK125R with ST6Gal1 knockdown, drug-resistant cell growth was higher on d15 on OP9 Galectin-1 KO cells (Figure 7d). We confirmed a relatively minor role for stromal Galectin-1 by comparing two other BCP-ALLs, LAX56 and LAX57, on OP9 Galectin-1 knockout and wild-type cells during a 12-day vincristine treatment. As shown in Appendix A, although the Galectin-1 knockout cells were moderately less effective in providing chemoprotection at day 3 of treatment, by day 12, this effect had disappeared and there was a trend towards better support by the Galectin-1 knockout stroma. Similar results were previously obtained using BCP-ALL cells overexpressing ST6Gal1, for which OP9 Galectin-1 KO cells provided comparable or better chemoprotection than wild-type OP9 cells [58].

We also investigated Galectin-1 and Galectin-3 content of the BCP-ALL cells. Under steady-state growth, BCP-ALLs that have migrated into the medium above the stromal layer normally express little if any endogenous Galectin-3 but vary in endogenous levels of Galectin-1 (Appendix A). Figure 8 shows JFK125 and JFK125R BCP-ALL cells from the OP9 Galectin-1 co-cultures collected at d15 for Western blot analysis. Consistent with our previous findings [27], compared to PBS-treated controls, vincristine treatment induced a notable increase in Galectin-3 levels in all eight samples. Galectin-1 signals in the same BCP-ALL samples showed little variation, regardless of whether they had been co-cultured with wt or Galectin-1 OP9 KO cells, suggesting most of the Galectin-1 was leukemia cell endogenous.

We also analyzed this in US7 cells, which express low levels of Galectin-1 ([28] and Appendix A). After 15 days of vincristine treatment, the signal for Galectin-1 in US7 cells grown on OP9 Galectin-1 knockout stromal cells was also similar to that of US7 grown on OP9 wt cells (Appendix A). Thus, the Galectin-1 signal in these BCP-ALL cells must be endogenously produced human Galectin-1. Consistent with our previous findings [25,26,27], we conclude that Galectin-3 levels are dynamically induced by chemotherapy. Here, we show that endogenous Galectin-1 levels do not appear to be similarly increased. Moreover, there is no evidence that Galectin-1 from stromal cells has a major contribution to the ability of BCP-ALL cells in evading the effects of chemotherapy. Finally, the reduction of α2-6 linked sialylation on N-glycans did not dramatically increase the levels of Galectin-3 or Galectin-1 associated with the BCP-ALL cells.

## 3. Discussion

### 3.1. Overlapping LGALS1 and LGALS3 mRNA Expression in BCP-ALL

There is abundant evidence that variations in both Galectin-1 and Galectin-3 levels correlate with changes that develop during leukemogenesis and its treatment. For example, total bone marrow samples of pediatric BCP-ALL patients treated with chemotherapy over the course of 33 days significantly increased expression of both *LGALS1* (Appendix A) and *LGALS3* (see [25]) mRNA. In such samples, it is difficult to ascribe the increased expression only to the leukemia cells, seeing that total bone marrow contains a variety of different cell types, of which the representation will change over the course of chemotherapy, and with differential expression of *LGALS1* and *LGALS3* in different hematopoietic cell types (Appendix A). However, Ebinger et al. [59] performed single-cell RNA sequencing on BCP-ALL cells, comparing dormant non-proliferating with actively dividing cells in the same population, and deriving a 250-gene signature characteristic of dormant leukemia cells that could give rise to relapse. *LGALS1* but not *LGALS3* is part of the dormant cell signature. The same authors also compared diagnosis to minimal residual disease samples after 33 days of chemotherapy, and in that data set, Galectin-3 but not Galectin-1 was significantly upregulated among 206 genes with differential expression. Indeed, minimal residual disease (MRD)-positive samples and relapse samples had higher *LGALS3* than MRD-negative or diagnosis samples [25]. Lower overall survival in a small adult BCP-ALL patient set was also associated with higher *LGALS1* expression (Appendix A), although this could have been caused by a higher representation of MLL-rearranged BCP-ALLs, which have both a worse prognosis as well as higher *LGALS1* expression compared to other subsets. Interestingly, higher *LGALS1* expression correlates with shorter disease-free survival in AML as well [52].

### 3.2. Expression of Galectin-1 and Galectin-3 May Be Differentially Regulated

Although such RNA expression studies report on the trends of endogenous Galectins, they have limitations because they may not capture actual protein levels in and on BCP-ALL cells. Indeed, in our studies, it was difficult to assign a specific contribution of Galectin-1 versus Galectin-3 because there are two potential sources of both extracellular/cell-surface-bound Galectin-1 as well as Galectin-3 protein: (1) that which is secreted by bone marrow stromal cells and can bind to the surface of the leukemia cells, and (2) that which is produced endogenously by the leukemia cells, of which an unknown fraction is transported to the cell surface and/or secreted. Similarly, intracellular Galectin-1 and/or Galectin-3 protein can have been synthesized endogenously or, alternatively, endocytosed from the surrounding extracellular space. For example, although endogenous Galectin-1 mRNA levels are higher in more primitive B-cell precursors, a specific stromal cell type has been identified in the bone marrow that also produces Galectin-1 and regulates early B-cell development [60], as further reviewed in [61]. Thus, the cellular origin of a specific Galectin protein found in and on BCP-ALL cells is difficult to tease out and may be subtype dependent. Nonetheless, our Western blot analysis (Figure 8 and Appendix A) shows that Galectin-1 found in the BCP-ALL cells tested here is mainly from endogenously synthesized origin. Thus, levels of BCP-ALL endogenous Galectin-1 protein may be determined more by the stage of early B-cell development and less by inflammation, whereas Galectin-3, at least in a tissue co-culture setting, is mainly stromal but can be induced endogenously by drug treatment. We also note as a caveat that the stromal cells used in our studies were not isolated from human patient samples. Such stromal cells would have undergone a coevolution with the developing leukemia mass in the bone marrow and could have a different Galectin expression pattern than the leukemia-naïve, mitotically inactivated stromal cells used here.

### 3.3. Overlapping Function of Galectin-1 and Galectin-3

Overlapping expression patterns of Galectin-1 and Galectin-3 raise the question of whether they have functional redundancy. This issue is important as it dictates whether one or both would need to be inhibited for an effective therapeutic strategy in cancer. Few studies have examined functional redundancy directly by genetic ablation of both Galectin-1 and Galectin-3. The original report describing the generation of double knockout mice documented that concurrent loss of *Lgals1* and *Lgals3* function is compatible with relatively normal development [56], indicating that these genes do not contribute critically to overall cell survival. Using double knockout cells, Clark et al. showed that Galectin-1 and Galectin-3 both regulate mature B-cell functions in a mouse model of autoimmune disease [62]. Moreover, Sirko et al. [63] showed that Galectin-1 and Galectin-3 together promote astrocyte reactivity, but that exogenously added Galectin-3 and not Galectin-1 can normalize the double mutant phenotype. In our study, we found that the double knockout Galectin-1 and Galectin-3 murine BCP-ALL cells proliferated significantly less well than wild-type controls, indicating that these cells clearly are defective in some aspects of mitogenic signaling, or have other endogenous deficiencies related to, for example, cell cycle progression. It is possible that the proliferation defect of the dKO BCP-ALL cells also caused the greater vulnerability to nilotinib or vincristine monotherapy seen in our study. Interestingly, the proliferation defects of the dKO BCP-ALL cells were somewhat mitigated by co-culture with a fibroblast stromal layer, and their ability to withstand drug treatment was also enhanced by the presence of these cells. Therefore, it is likely that the stromal layer is providing some of the missing functions, which could include the production and secretion of Galectin-3 [27].

On a protein interaction level, Galectin-1 and Galectin-3 have many binding partners in common, such as the von Willebrand factor, the lysosomal protein Lamp1 [64,65,66], and the mature B-cell transcription factor OCA-B [67]. In human mesenchymal retinal pigment epithelial cells, many other glycoproteins were identified that bind both Galectin-1 and Galectin-3 [68]. In agreement with functional overlap, simultaneous knockout of Galectin-1 and Galectin-3 in MEFs strongly reduced EGF-stimulated activation of K-Ras, Erk, and Akt, compared to Galectin-1 or Galectin-3 single null mutant MEFs [69]. In our studies, using a different cell type, we did not detect differences in baseline levels of activated Erk or Akt in *Lgals1 × Lgals3 -/-* dKO BCP-ALL cells compared to wt. This could be explained because we studied cancer cells, which express Bcr/Abl as driver oncogene, obviating the need for exogenous cytokine signaling and making the cells IL-7 independent, while activating multiple endogenous signal transduction pathways [57]. Interestingly, we did detect increased phosphorylation of an activating tyrosine residue common to a number of Src-family kinases (SFK) such as Lck and Lyn, in the double knockout cells. The antibodies used here are directed against Y419 in Src but may also react with Y394 in Lck, or Y397 in Lyn, both of which are also highly expressed in BCP-ALL. This residue can be dephosphorylated by the transmembrane phosphatase CD45 [70,71]. Although it will require further investigation, the specific loss of Galectin-1 could be the cause of increased SFK phosphorylation since Galectin-1 is known to regulate the basal and activating signaling of such SFK through binding to the extracellular domain of CD45 and regulating its phosphatase activity towards SFK, as reported in T cells [72,73].

### 3.4. A Common Function for Galectin-1 and Galectin-3 in Regulating BCP-ALL Cell Migration?

We found that Galectin-1 and Galectin-3 both regulate BCP-ALL migration (this study and [25,28]). This may be a common mechanism through which Galectin inhibition causes chemo-sensitization. GM-CT-01 and GR-MD-02 both inhibited human BCP-ALL migration, presumably by interfering with a signal produced by engagement of extracellular Galectin-3 and possibly Galectin-1 with an, as of yet unknown, receptor or alternatively, inhibit Galectin internalization and by this mechanism inhibit intracellular effects. In support for a contribution of intracellular Galectins, we here found that endogenous Galectins (either Galectin-1, Galectin-3, or both) also regulate motility because *Lgals1 × Lgals3 -/-* dKO BCP-ALL cells had reduced migration to stromal fibroblasts. However, because we were not able to knock out either Galectin-3 or Galectin-1 in human BCP-ALL cells (not shown), the specific contribution of this source of Galectin, if any, to human BCP-ALL migration remains unexplored.

Mechanistically, an intracellular contribution of Galectin-3 to cell migration could be inferred from its association with non-glycosylated cytoplasmic proteins involved in cell motility, whereas this is less clear for Galectin-1. Galectin-3 was reported to form a complex with the actin-binding tyrosine kinase c-Abl [74,75,76], and we were able to also co-immunoprecipitate Galectin-3 with c-Abl and Bcr/Abl (Appendix A). The non-muscle myosin MYH9/NMIIA could also contribute to regulating motility. Nakajima et al. [77] reported that Galectin-3 interacts with myosin IIA and regulates cellular morphology during bone remodeling and osteoclast maturation associated with metastasis. We also found a physical association between Galectin-3 and MYH9 in BCP-ALL cells (Appendix A). Moreover, chemotaxis of BCP-ALL cells toward supportive stroma and SDF-1α is, in part, regulated by MYH9 because treatment of BCP-ALL cells with blebbistatin significantly decreased their adhesion and migration (Appendix A). This suggests that upon entering BCP-ALL cells, extracellular Galectin-3 can bind with MYH9, a component of the motility machinery in these cells, and in doing so may positively regulate multi-protein complexes needed for leukemic cell adhesion and migration.

### 3.5. Contribution of Extracellular Galectins in the Microenvironment of BCP-ALL Cells Sourced from Bone Marrow Stromal Cells

Extracellular Galectins of both endogenous secreted and stromal secreted origins bind to poly-LacNAc-modified glycoproteins on the cell surface, which results in glycoprotein lattice formation and receptor clustering. Since numerous studies reported an inverse functional connection between ST6Gal1-mediated sialylation of N-glycans and the binding of Galectin-1/-3 [78,79,80,81] (Appendix A), we investigated if drug resistance development was regulated by the combination of ST6Gal1 activity and extracellular, stromal-produced Galectin-1 or -3. However, in BCP-ALL cells, we did not find evidence for a critical regulatory effect of ST6Gal1 on chemoprotection mediated by stromal-produced Galectins-1/-3. Thus, it is possible that Galectin-1-/3 bind to critical O-glycans, which are not substrates for ST6Gal1 [82,83,84], or to critical N-glycans that are not ST6Gal1 substrates in BCP-ALL cells. Finally, because ST6Gal1 sialylation of N-glycans inhibits Galectin-1 binding but only some Galectin-3 binding [2], and we did not find evidence for a protective effect on BCP-ALL cells for extracellular stromal Galectin-1, this may simply reflect a greater contribution of extracellular Galectin-3. However, since the small molecule Galectin-1 inhibitor PXT008 was able to sensitize BCP-ALL cells to vincristine [28], endogenous Galectin-1 still appears to be of major significance for BCP-ALL cells that express it in high quantities.

### 3.6. Prospects for Dual Galectin-1/-3 Inhibition for Treatment of BCP-ALL

We conclude that a strategy of combined inhibition of both Galectin-1 and Galectin-3 function, both extracellularly and intracellularly, using small molecule inhibitors, would seem to be particularly effective to chemo-sensitize BCP-ALL cells. We used GR-MD-02 and GM-CT-01 as putative dual Galectin-1/Galectin-3 inhibitors on the basis of in vitro data, reporting a similar K_d_ for GM-CT-01 and GR-GM on Galectin-1 and Galectin-3 [48]. However both compounds have been used as if they mainly inhibit extracellular Galectin-3 (GM-CT-01 [85]; GR-MD-02 [86,87,88,89]). GM-CT-01 interacts with a Galectin-1 domain not including the classical carbohydrate-binding site, but this is in vitro [45]. Moreover, Stegmayr et al. [90] reported an IC_50_ of 4 mg/mL for Galectin-3 and >20 mg/mL for Galectin-1 using GM-CT-01 in an in vitro inhibitory binding assay to asialofetuin. These high concentrations are in agreement with the amounts of GM-CT-01 needed to obtain a biological response in our studies and raise the concern for off-target effects. Thus, a need remains for the development of more potent Galectin-1 inhibitors as well as for additional anti-Galectin-3 drugs. Novel potential therapeutics include, among others, anti-Galectin-3 antibodies [91], taloside-based carbohydrate mimetics [25,92], selenylated thiodigalactose analogs [93], heparin-based Galectin-3 inhibitors [94], and inhibitory peptides [95]. Finally, although the high concentrations of Galectin-1 and -3 found in stromal and leukemia cells may impair our ability to completely inhibit them pharmacologically, our results suggest that such inhibitors could still be extremely useful, and probably not toxic, when combined with other therapies in both leukemias, and possibly other cancers, as reviewed in [51].

## 4. Materials and Methods

### 4.1. Gene Expression Analysis

Meta-analysis of *LGALS3* and *LGALS1* (Galectin-3 and Galectin-1) expression of RNAs from pediatric Ph-positive ALL at diagnosis was performed on GEO Datasets accession GSE28497 and GSE79533, described in [38] and in [39], respectively. Processed data in the series matrix files represent values normalized by MAS5.0 and baseline transformed to a median target intensity. Txt file values for *LGALS3* and *LGALS1* (probe sets 208949_s_at and 201105_at) imported into Excel were manually extracted into Prism5.0.

Mice transgenic for the human P190 form of Bcr/Abl (Jackson Labs strain 017833, Bar Harbor, ME, USA) develop precursor B-lineage (pre-B) acute lymphoblastic leukemia, on average within 3 months of birth, when on a C57Bl/6J background. Bone marrows were isolated from control C57BLl/6J mice and from transgenic Bcr/Abl mice when they had not yet developed full-blown leukemia (<60 days of age), from Bcr/Abl transgenic mice with overt leukemia and packed bone marrows (>90 days of age), and from fully leukemic mice that had received a seven-day treatment with 75 mg/kg AMN107 (nilotinib; Novartis Basel, Switzerland). Three mice were used per condition, and cells from each mouse were processed separately. Pre-B-cells from these twelve bone marrows were flow-sorted using CD19 and AA4.1 as markers. Total RNA from CD19+ AA4.1+ cells used for microarray analysis was isolated by RNeasy (QIAGEN, Germantown, MD, USA)) purification. Double-strand complementary DNA was generated from 5 µg of total RNA using a poly(dT) oligonucleotide that contains a T7 RNA polymerase initiation site and the SuperScript III reverse transcription (Invitrogen, Waltham, MA, USA). Biotinylated cRNA was generated and fragmented according to the Affymetrix protocol and hybridized to 430 mouse microarrays (Affymetrix, Santa Clara, CA, USA). As described in Trageser et al. [42], Cel files from GeneChip arrays were imported to the BRB Array Tool (http://linus.nci.nih.gov/BRB-ArrayTools.html, accessed 3 March 2009) and processed using the RMA algorithm (Robust Multi-Array Average) for normalization and summarization. Data were deposited in GEO as GSE110104.

### 4.2. Treatment with Drugs, Cell Proliferation and Viability, and Flow Cytometry

GM-CT-01 and GR-MD-02 were provided by Galectin Therapeutics, Inc. (Norcross, GA, USA) and were stored at 4 °C. Nilotinib (AMN107) was obtained from Novartis (Basel, Switzerland). Nilotinib was dissolved in DMSO and stored at −20 °C. A vincristine sulfate solution was obtained from Hospira Worldwide Inc. (Lake Forest, IL, USA). 

For proliferation assays, US7 or TXL2 cells were cultured in a 96-well plate at a density of 5 × 10^4^/well in the presence of irradiated OP9 cells. Cells were treated with GM-CT-01 or with GR-MD-02, in combination with nilotinib or vincristine. Controls for nilotinib or vincristine were DMSO at the dilution matching the drug samples. After drug exposure, cells were collected and re-suspended in culture medium containing 0.1% (wt/vol) Trypan blue. Trypan-blue-excluding and total cells were counted using a hemocytometer. All drug sensitivity assays were performed in triplicate wells. Viability of the cells is expressed as the percentage of Trypan-blue-excluding cells of the total number of cells. Data points show the mean ± SEM of triplicate samples. Cell counts were also performed using a CellTiter-Glo luminescent cell viability assay (Promega, Madison, WI, USA) where indicated on cells migrated into the culture medium.

To assay the ability of GM-CT-01 to displace Galectin-3 using FACS, TXL2 cells were harvested from underneath the OP9 stromal layer. Cells were incubated with or without 20 mg/mL GM-CT-01 for 2 h at 37 °C in complete medium, then washed with PBS. DTSSP was added at 5 mM to cross-link extracellular Galectin-3 to the cell surface. After a 30 min incubation at RT, the reaction was terminated by a 15 min incubation with 50 mM glycine at pH 7.5. After a wash in PBS, cells were incubated with PE-Galectin-3 (Biolegend cat# 126706, San Diego, CA, USA) or matched isotype control for 15 min and analyzed on BD Accuri C6 cytometer (BD Biosciences, San Jose, CA, USA).

FACS analysis of mouse wt and dKO BP ALL cells used antibodies against AA4.1 (eBiosciences cat#17-5892-82, San Diego, CA, USA), CD43 (BD cat#553270, San Jose, CA, USA), and CD45/B220 (Biolegend cat#103132, San Diego, CA, USA). Gates were set on the basis of isotype controls.

### 4.3. Migration Assay

For migration assays, human TXL2 or US7 cells (1 × 10^5^) were seeded into the upper well of Transwell plates with a 5 μm pore size. The lower chamber contained a layer of irradiated OP9 stromal cells with different concentrations of GR-MD-02 or GM-CT-01 as indicated. Wells without stromal cells in the bottom chamber served as controls. ALL cells migrated to the bottom wells were counted after overnight incubation. Migration of wt1, wt2, dKO1, and dKO2 BCP-ALL cells was measured individually.

### 4.4. Cells and Animal Use

*Animal use:* Use of mice for harvest of MEFs and leukemia cells was approved by the Children’s Hospital of Los Angeles IACUC protocol #24-14.

*Murine BCP leukemia:* Murine leukemia cells were generated from bone marrows of age- and sex-matched wild-type (129 P3/J *Lgals1 × Lgals3 ^+/+^*) controls and Galectin-1/Galectin-3 (129 P3/J *Lgals1 × Lgals3*
^−/−^) double knockout mice on a C57Bl/6J background. Bone marrow cells were transduced with p190 Bcr/Abl-encoding retroviruses as previously described [23]. In brief, transduced bone marrows were grown for 5 days with 10 ng/mL rmIL-7 and cultured in IMDM with 50 μM β-mercaptoethanol in 20% FBS, 1% L-glutamine, and 1% penicillin/streptomycin. The murine BCP-ALL cells were grown without MEFs except where indicated. All assays were performed within ≈6 months of the initial transduction.

*Cell lines and PDX-derived human BCP-ALLs:* The murine OP9 stromal cell line (CRL-2749) and the human CML cell line K562 were obtained from the ATCC (Manassas, VA, USA). Human ALL cells including Ph-positive TXL2, Ph-negative US7 cells, and Ph-like P2RY8-CRLF2 fusion-positive JFK125 and JFK125R were described previously [22,96]. Human leukemia cells were grown in MEM-α medium supplemented with 20% FBS, 1% L-glutamine, and 1% penicillin/streptomycin (Invitrogen Corporation, Waltham, MA, USA). Human BCP-ALL cells were cultured in the wells of a 6-well plate or a 96-well plate at a density of 0.5–1 × 10^6^ cells/mL, in the presence of irradiated or mitomycin C-treated OP9 cells, as described [58].

*Mouse embryonic fibroblasts (MEFs)*: E13.5 embryos were isolated from timed matings of C57Bl/6J mice. Internal organs and heads were removed, and tissue was homogenized by mincing with razor blades and pressure from the back of a syringe [97,98,99]. Single-cell suspensions were further generated by a 30 min incubation at 37 °C in 5 mL Trypsin-EDTA. Tissue chunks were removed with a cell strainer, and cells were plated at a density of 6 × 10^6^ per 15 cm dish in DMEM + 10% FBS, P/S, and L-glutamine.

### 4.5. Western Blotting and Immunoprecipitation

For detection of NF-kB (p100/52, p65), a nuclear extraction kit (Imgenex, San Diego, CA, USA) was used to separate nuclear and cytoplasmic fractions. Cell extracts were subjected to 8–15% sodium dodecyl sulfate–polyacrylamide gel electrophoresis. The antibodies used include pY20 (BD-Transduction, San Jose, CA, USA), Galectin-3 (rat, Biolegend, San Diego, CA, USA), phospho-ERK1/2, phospho-p38, phospho-STAT5, phospho-AKT, AKT (Cell Signaling Technology, Danvers, MA, USA), ERK1/2, NF-kB p65, Bcr N-20 (Santa Cruz Biotechnology, Santa Cruz, CA, USA), NF-kB p100/52 (Millipore, Burlington, MA, USA), Galectin-1, and ST6Gal1 (R&D Minneapolis, MN, USA) using standard procedures. Gapdh (Chemicon International, Rolling Meadows, IL, USA) or β-actin (Sigma, Santa Cruz, CA, USA) antibodies were used as a loading control. The phospho-SFK (Y416, Cell Signaling Technology, Danvers, MA, USA) antibody recognizes pY416 in Src but may cross-react with other Src family members (Lyn, Fyn, Lck, Yes, and Hck) when phosphorylated at equivalent sites. For Galectin-1 and Galectin-3 Western blot analysis (Figure 8 and Appendix A), after 15 days of culture, human BCP-ALL cells in the supernatant and attached to the stromal layer were collected. OP9 cells had been mitotically inactivated by mitomycin C treatment before the start of the culture at d0. Cells were passed through a 35 µm nylon mesh cell strainer before lysis in Triton-X100 lysis buffer (Thermo-Fisher, Alfa Aesar, Stoughton, MA, USA).

### 4.6. Statistical Analysis

Statistical tests are mentioned in the figure legends. A false discovery rate approach was adopted to decide the significance of the comparison on the basis of the fact that no more than 5% of the discoveries will be false discoveries. Pearson correlation analysis was performed to examine expression correlations between *LGALS1* and *LGALS3* genes in public data sets. A two-tailed *p*-value was calculated to suggest the significance of the correlation, and a *p*-value < 0.05 was used to decide whether the two sets of expression values were correlated.

## Figures and Tables

**Figure 1 ijms-23-14359-f001:**
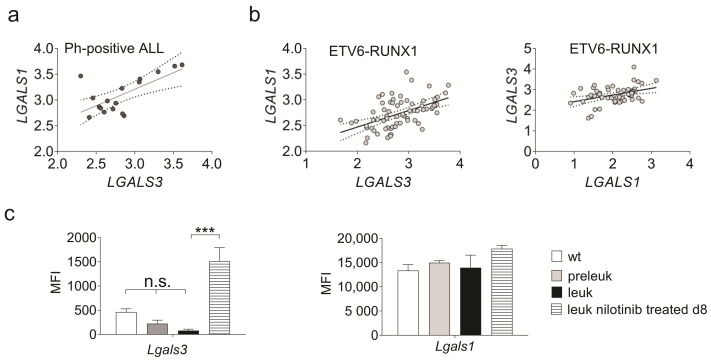
*LGALS1* and *LGALS3* mRNAs are co-expressed in human and mouse BCP-ALL cells. Positive correlation between log10 *LGALS1* and *LGALS3* mRNA expression in (**a**) human Ph-positive ALL samples (GSE79533) and (**b**) ETV6-RUNX1 ALL samples in two different arrays (left, GSE79533; right, GSE28497). Pearson correlation analysis, where dotted lines indicate the 95% confidence range of the best fit line. (**c**) Meta-analysis of GSE110104 for Galectin-3 (*Lgals3*) and Galectin-1 (*Lgals1*) gene expression in mouse B-cell precursor cells, flow sorted from bone marrows of control wild-type (wt) mice and pre-leukemic; fully leukemic (leuk); and leukemic, 75 mg/kg/d d8 nilotinib-treated *BCR/ABL* P190 transgenic mice. Each set consists of three biological replicates. *** *p* < 0.001. One-way ANOVA, Tukey’s multiple comparisons test. Right panel, no significant differences. MFI, mean fluorescent intensity.

**Figure 2 ijms-23-14359-f002:**
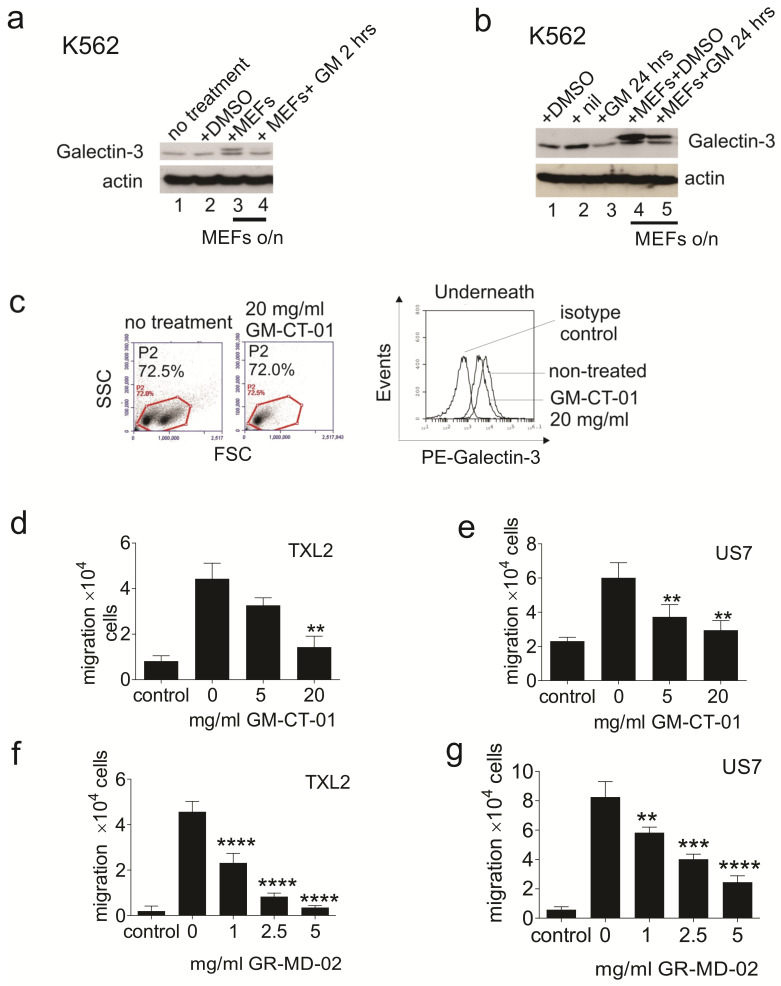
Interference of interaction of extracellular Galectin with leukemia cells by large inhibitory carbohydrates. (**a**,**b**) Western blot analysis of Galectin expression in K562 cells grown in suspension or co-cultured overnight with MEFs. Actin—loading control. (**a**) Treatment with 5 mg/mL GM-CT-01 (GM) for 2 h. (**b**) Treatment of cells with 1 μM nilotinib (nil) or 2 mg/mL GM-CT-01 for 24 h. (**c**) BCP-ALL cells in co-culture with OP9 cells were treated with 20 mg/mL GM-CT-01 for 2 h. BCP-ALL cells from underneath the stroma were assayed for cell surface expression of Galectin-3 using FACS: left, gating; right Galectin-3 cell-surface-positive cells in gate P2. Control, BCP-ALL cells underneath OP9 not treated with GM-CT-01. (**d**–**g**) Overnight migration of human BCP-ALL TXL2 (Ph-positive) or US7 (Ph-negative) to OP9 stromal cells plated in the bottom of Transwells in the absence or presence of the indicated concentrations of GM-CT-01 (**d**,**e**) or GR-MD-02 (**f**,**g**) ** *p* < 0.01, *** *p* < 0.001, **** *p* < 0.0001, one-way ANOVA, between 0 mg/mL compound and samples incubated with the indicated amounts of compound. Controls: only medium, no OP9 cells in the bottom well.

**Figure 3 ijms-23-14359-f003:**
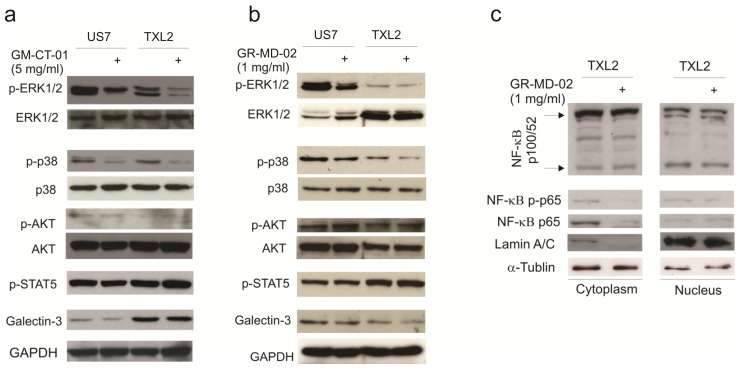
GM-CT-01 and GR-MD-02 attenuate mitogenic signaling in ALL cells. (**a**) Western blot analysis of US7 and TXL2 total cell lysates after incubation with 5 mg/mL GM-CT-01 for 72 h in the presence of OP9 stromal cells. After pErk1/2 antibody exposure, blots were stripped and re-probed. (**b**) US7 and TXL2 cells were incubated with 1 mg/mL GR-MD-02 for 72 h in the presence of OP9 cells. Subsequent to pErk1/2 immunoblotting, blots were stripped and reprobed. Gapdh, loading control. (**c**) NF-κB p100/52, RelA NF-κB p-p65 and NF-κB p65 WB on TXL2 cytoplasmic and nuclear fractions. Lamin A/C and α-tubulin, loading controls for nuclear and cytoplasmic fractions, respectively.

**Figure 4 ijms-23-14359-f004:**
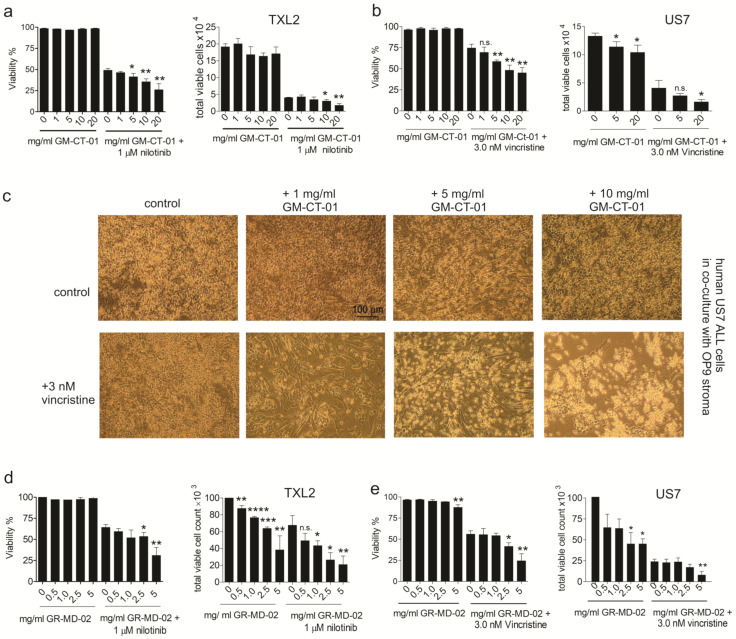
Combination drug treatment with GM-CT-01 or with GR-MD-02 inhibited proliferation of ALL cells. (**a**) Viability (left panel) and cell counts (right panel) of Ph-positive TXL-2 BCP-ALL cells treated with GM-CT-01 and nilotinib for 96 h in the presence of OP9 cells. (**b**) Viability (left panel) and cell counts (right panel) of US7 cells treated for 96 h with GM-CT-01 alone or in combination with vincristine in the presence of OP9 cells. (**c**) Phase contrast images of US7 cells co-cultured with irradiated OP9 stroma and treated with GM-CT-01 and vincristine as indicated. (**d**,**e**) Viability and cell counts of TXL2 (**d**) or US7 (**e**) treated with GR-MD-02 or a combination treatment as indicated. Viable cells and cell numbers were determined using Trypan blue exclusion. * *p* < 0.05, ** *p* < 0.01, *** *p* < 0.01, **** *p* < 0.0001 compared to 0 mg/mL samples, representing treatment with nothing or only with nilotinib or vincristine, unpaired *t*-test. Control samples, DMSO at the same final concentration as the vincristine/nilotinib samples.

**Figure 5 ijms-23-14359-f005:**
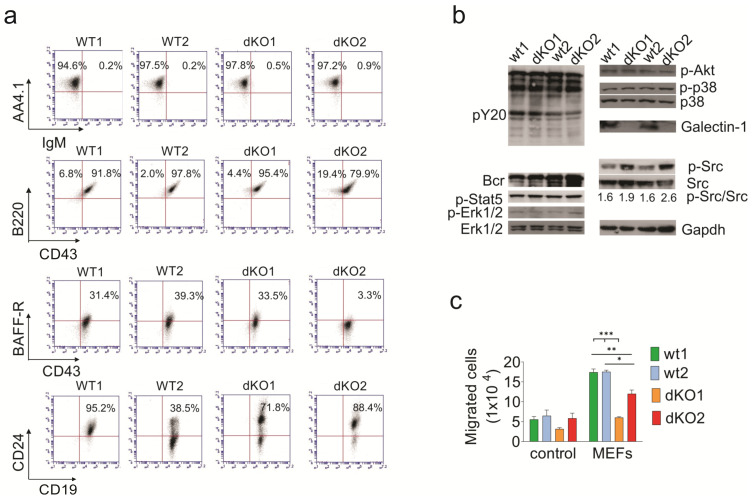
Characterization of murine Bcr/Abl-positive BCP-ALL cells lacking Galectin-1 and Galectin-3 generated by transformation of *Lgals1 × Lgals3 -/-* or matched wt bone marrow cells with p190 Bcr/Abl. Samples include wild-type (wt) pre-B ALL #1 and #2, and *Lgals3 × Lgals1-/-* (dKO) pre-B ALLs dKO1 and dKO2, representing independent transductions. (**a**) FACS analysis for the indicated markers at week 3 after transduction. Gates were set using isotype controls. Numbers: % of cells in indicated quadrants. (**b**) Western blot analysis with the antibodies indicated to the sides of the panels. The ratio of pSrc/Src was determined by densitometric scanning of Western blot film images using ImageJ software version 1.51. Gapdh, loading control. BCP-ALL cells grown without stroma. (**c**) Mouse BCP-ALL wt1/wt2 and dKO1/dKO2 cells were plated in the upper wells of a Transwell and allowed to migrate overnight towards control medium or MEFs plated in the bottom well. * *p* < 0.05, ** *p* < 0.01, *** *p* < 0.001 for differences between wt and dKO samples migrating toward MEFs (n = 3 per sample per genotype), unpaired two-tailed *t*-test.

**Figure 6 ijms-23-14359-f006:**
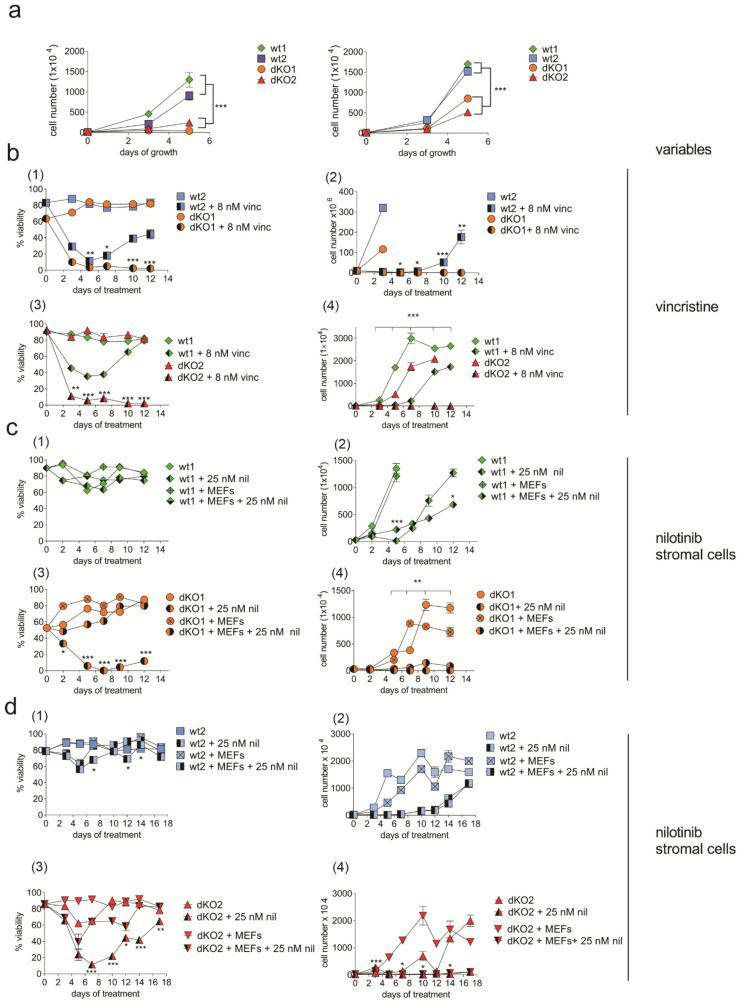
Murine Bcr/Abl-positive BCP-ALL cells lacking Galectin-1 and Galectin-3 have reduced growth and survival. (**a**) Proliferation as measured by viable cell counts of the indicated genotypes. Left panel, 7 weeks after transduction; right panel, 8 weeks after transduction. *** *p* < 0.001 for the indicated group comparisons. (**b**–**d**) Comparative analysis of viability (left panels) and proliferation (right panels) of dKO and wt ALL cells in a long-term drug treatment. (**b**) Cells treated with vincristine. Values for dKO + 8 nM vincristine compared to wt + 8 nM vincristine: * *p* < 0.05, ** *p* < 0.01, *** *p* < 0.001. (**c**,**d**). Nilotinib treatment. Experiments with wt ((1) and (2)) and dKO ((3) and (4)) cells grouped in (**c**,**d**) were performed together but are shown in separate graphs for clarity. Cells at each time point grown without and with MEFs while treated with nilotinib are compared pairwise at each time point for statistically significant differences. Significant differences are indicated. * *p* < 0.05, ** *p* < 0.01, *** *p* < 0.001. Two-way ANOVA, multiple comparisons.

**Figure 7 ijms-23-14359-f007:**
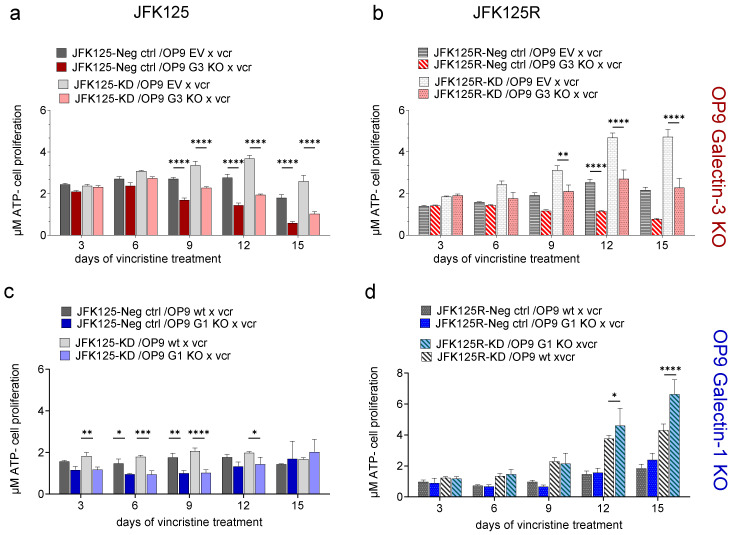
Reduced growth of human BCP-ALL leukemia cells in co-culture with OP9 Galectin-3 knockout but not Galectin-1 knockout cells when treated with 1.5 nM vincristine chemotherapy. ATP levels measured against a standard curve (CellTiterGlo assay, Promega). (**a**,**c**) JFK125 *ST6GAL1* knockdown or negative control cells or (**b**,**d**) JFK125R *ST6GAL1* knockdown or negative control cells [58] co-cultured for 15 days with OP9 Galectin-1 (G1) or Galectin-3 (G3) knockout cells. Two-way ANOVA, pair-wise comparisons only between matched BCP-ALLs on OP9 control versus OP9 KO cells. N = 4 samples per time point. Data for JFK125R NegC and KD on OP9 wt (NegC OP9) are from Zhang et al. [58] and are shown for comparison. Two-way ANOVA, Tukey’s multiple comparison test, adjusted *p*-values. * *p* < 0.05, ** *p* < 0.01, *** *p* < 0.001, **** *p* < 0.0001.

**Figure 8 ijms-23-14359-f008:**
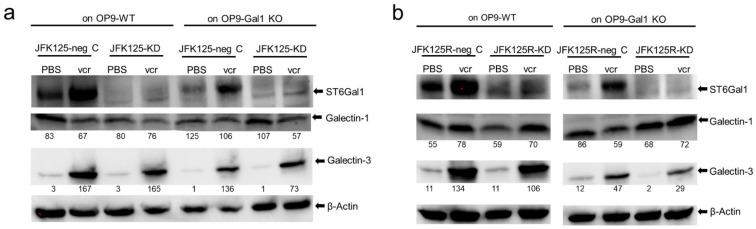
Drug-resistant JFK125 and JFK125R cells contain constant Galectin-1 with increased Galectin-3. JFK125 (**a**) or JFK125R (**b**) cells were harvested from co-cultures with OP9 wild-type (WT) or Galectin-1 knockout (KO) cells as indicated after treatment for 14 days with 1.5 nM vincristine or PBS. Antibodies used for Western blotting are indicated to the right and include ST6Gal1 (1:500), Galectin-3 (1:1000), Galectin-1 (1:1000), and β-actin (1:500, Santa Cruz); 4–20% SDS-PAA gradient gel; 1.6 × 10^6^ cell equivalents loaded per well. Numbers below the Galectin-1 and Galectin-3 panels indicate the percentage of β-actin loading control signal in each lane by ImageJ quantification of shorter chemiluminescent exposures of the same Western blots.

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
