# Peer review of "Galectin-1 and Galectin-3 in B-Cell Precursor Acute Lymphoblastic Leukemia"

_ijms, 2022, doi:10.3390/ijms232214359_

Round 1

Reviewer 1 Report

In their manuscript, Fei et al define the effects of Galectin-1 and Galectin-3 produced by BCP-ALL and stromal cells on BCP-ALL fitness using leuekmia and stromal cells from galectin knockout mice. The authors examine BCP-ALL survival, migration and chemotherapy response. Authors used different cell types Galectin-1 and Galectin-3 knockouts in addition to Galectin-1 and Galectin-3 inhibitors to explore the role of the different sources in the tumor microenvironment on the tumor behavior. Establishing the role of Galectin-1 and Galectin-3 can support the development of specific inhibitors as potential stand-alone or combination therapeutics, so these findings will be of interest to leukemia researchers. However, there are some problems with the order in which data are presented that make the manuscript difficult to follow and lessen the overall impact of the author’s conclusions. Below are some opportunities for improvement across the manuscript

Major:

1.      The authors state that no mice were used in these studies but methods include timed matings and embryo harvests for MEFs, which involve sacrifice of the mother. Leukemias were generated from bone marrows of age and sex matched mice. This work does use animals, and cannot be published if there were no institutional approvals to do these experiments.  

2.      The introduction lacks sufficient general background on Galectins, which would be useful for readers not familiar with these proteins. The flow of the introduction could also be revised to strengthen the manuscript. For example, the first paragraph describes  Ph-postive ALL and the related model without providing a clear link to the paper’s scope of Galectin-1 and Galectin-3 role in this type of ALL. In addition, the introduction didn’t provide the informatipn needed to highlight what the impact of elucidating the combined role of Galectin-1 and Galectin-3 in BCP-ALL would be. Perhaps the authors would consider adding a paragraph on the potential impact of studying the role of combined role Galectin-1 and Galectin-3 in providing more insights toward BCP-ALL microenvironment, cell-cell interaction, and metastasis.

3.      In section 2.1, normal and abnormal hematopoietic cell types are examined but stromal cells are from normal human bone marrow or OP9 bone marrow stromal cells. Comparison of stromal cells from a leukemic tumor microenvironment would strengthen conclusions.

4.      Section 2.1: Paragraph 4, please provide more context why gene expression was evaluated after  Imatinib treatment. Why did the authors decide to treat with Imatinib rather than nilotinib? If this was a treatment regimen, both the stromal cells and the BCP-ALL will be under the effect of the same drug at a given time point, and the authors mentioned previously that nilotinib is the preferred treatment for Ph+ BCP-ALL.

5.      Where did the Lgals1 x Lglas3 -/- mice come from or how were they made? There is no reference for them at first use in the paper or in the methods.

6.      In figure 2a: Last FACS set, how you would explain the different CD24 and CD19 expression on the two WT cells—one had 95.2% of cells CD24+/CD19+, while the other was only 38.5%.  

7.      Figure 2C, the Galectin-3 western blot is aggressively cropped—please show more of this blot, it is difficult to see the bands with such tight cropping. As it is, it seems as though the dKO1 cells have expression of Galectin-3 as there is a clear band darker in appearance than the WT groups. The authors even state that 3 of the 4 lines have detectable Galectin 3. The one line it is not detetable in is the WT. How is this possible if the cells are from a knockout line? Please explain.

8.      Section 2.2: Since there was no clear and consistent effect on Galectin-3 expression as evidenced by the western blot in figure 2c, please explain how this section correlates the physiological effect to the knock out of both genes and not as a result of the knock out of one of them?

9.      Section 2.2: Paragraph 6 and 7, please explain the rationale for why two different chemotherapy drugs (vincristine and nilotinib) were used in evaluating proliferation and stromal cell effect respectively and not only one of them?

10.   Section 2.3 and 2.4: I think there is flow problems preventing the delivery of a coherent story. So authors started highlighting that Galectin-1 and Galectin-3 are co-expressed in BCP-ALL, then moved to the characterization of the dKO model and the physiological effects of the dKO. After that, authors went back to the effect of exogenous Galectin on migration. If section 2.3 were before the dKO section, it may better set the stage for the importance of Galectin in tumor behavior, and then move to the proof of concept via the dKO.

Minor

1.      Section 2.1: Line (111) please site the data set that demonstrates the positive correlation.

2.      Typo in Section 2.1: Line (121) fully leukemia should be fully leukemic.

3.      Section 2.1: Line (130) Please define MSC in the list of abbreviations.

4.      Figure 1c: Please use the common legend to the most right to be consistent with the rest of the figures.

5.      Figure 2a and other figures showing FACS plots: the numbers in the FACS plots along the X and Y axis and the red text in the quadrants are too small to be readable in the figure. Either remove them or make them larger if they are important to understand the figure.

6.      Figure 2d: why are WT and dKO grouped, while in others they were WT1, WT2, dKO1 and dKO2? Please stay consistent and show the data from the individual sets.

7.      Figure 3: To make the graphs easier for the reader to interpret, please consider using different colors to indicate wt and dKO. Use consistent legends for wt and dKO across (a-d) for the reader to be able to navigate through the figures without needing to revisit the legend every time. Finally, be consistent with the order of legend between part c and part d. For part a, right panel, use days of growth on the x-axis instead of days to match the left panel.

8.      Figure 4: Part a, include loading control.

9.      Figure 4: Part (a-c) please describe why lactose was included?

10.   Figure 4: Part b, please label the first two lanes. Also, why is the densitometry shown in this panel but not others and no numbers given? It is not clear what the authors are wanting to show with this information.

11.   Figure 5a lacks a loading control.

12.   Figure 6: Part c, the size of the images is too small for the reader to see the difference. Please also describe the result more technically.

13.   Figure 7: the font appears different than the rest of the figures. Please correct to keep consistent. 

Reviewer 2 Report

Fei F et al. presented a wide research on the role of Galectin-1 and Galectin-3 in B-cell precursor Acute Lymphoblastic Leukemia.

The study is well designed and presented in details. 

However there are some comments to offer:

1-) Some part of the article are redundant: please provide to express clearly the concept in few words.

2-) Row 113 and Figure 1b. Please substitute Tel-AML1 definition with ETV6::RUNX1. 

3-) 2.4 Section. Figure 6. In order to better understand the efficacy of the combination drug treatment with GM-CT01 or with GR-MD02, please provide the results of cell-lines (TXL2 and US7) exposure to Nilotinib and Vincristine alone. 

4-) Row 425. Please provide a better definition: MLL is a mixed lineage leukemia gene. Mixed Lineage Leukemias is a rare immunophenotypic subtype of acute leukemias. I am wondering if authors intended ALL with MLL rearrangements. The message is unclear.  

Reviewer 3 Report

The authors present data that suggests likely roles for Galectin-3 and/or Galectin-1 in B-ALL resistance to therapy. While much of the data is of some interest, the manuscript is difficult to follow and the data presented does not completely support the conclusions made.

Specific comments: 

1) Referring to "ST6Gal1 knockdown" cells in abstract adds immediate confusion for readers unfamiliar with the space - there is no description of what these are or why they're used.

2) Figure 2: Why is increase in Gal-3 following co-culture with stroma shown while this is not looked at for Galectin-1? In Fig 2c there is no band indicating Gal-3 expression in wt2 sample.

3) Pg 6 line 206 - description of MEFs should be moved up to first mention (on line 189)

4) Regarding Fig 4: While it is stated that treatment with GM-CT-01 had no effect on Gal-1 levels (pg 8 line 249), this is not actually shown in presented data. The WB does not appear to be able to differentiate between mouse and human Gal-1 (as is seen for Gal-3) and there is no flow done to show that Gal-1 is not affected (as per Fig 4d for Gal-3). It is also unclear why the flow plots shown (SSC v FSC) in Fig 4d are so different between treated and untreated samples i.e. there are 2 separate populations of cells in the untreated samples, compared to 1 in the treated samples.

5) Pg 15 line 495-496: How can the conclusion that Gal-1 and Gal-3 both regulate BCP-ALL migration based on data using a double knockout cell line?

Round 2

Reviewer 3 Report

The clarity of the manuscript and the overall message has been improved by some re-ordering of results. However, there remains some concerns regarding some data that was initially confusing and now has been entirely removed from the manuscript. 

Specifically:

1) the data pertaining to the  new Figure 2 now has no mention at all of Gal1. The original WB (now removed) was unable to show any changes to intra- or -extra-cellular Gal1 (as is shown for Gal3) and no flow data is provided looking at Gal1 levels (as for Gal3). While Gal1 is not specifically mentioned in results and authors use the term "interference with Galectin binding" generally, given there is no evidence provided for a role for Gal1 in these processes at all, it is then unclear why a double knockout is used rather than a Gal3 only k/o (or at least a combination of both).

2) The removal of the original Fig 2c (now Fig 5) means there is now no confirmation that Gal3 is knocked out in the double ko cells. And having seen this figure initially there remains the question why WT2 cells do not appear to have any Gal3 expression. 

Author Response

The clarity of the manuscript and the overall message has been improved by some re-ordering of results. However, there remains some concerns regarding some data that was initially confusing and now has been entirely removed from the manuscript.

Specifically:

1) the data pertaining to the new Figure 2 now has no mention at all of Gal1. The original WB (now removed) was unable to show any changes to intra- or -extra-cellular Gal1 (as is shown for Gal3) and no flow data is provided looking at Gal1 levels (as for Gal3). While Gal1 is not specifically mentioned in results and authors use the term "interference with Galectin binding" generally, given there is no evidence provided for a role for Gal1 in these processes at all, it is then unclear why a double knockout is used rather than a Gal3 only k/o (or at least a combination of both).

Response: The reviewer’s reasoning and comments were not entirely clear to us. The dKO cells were primarily intended to examine the contributions of endogenous Galectin-1/-3 whereas the experiments with the carbohydrate inhibitors (Figure 2) were looking at extracellular/cell surface Galectins. In the original submission we showed the dKO data followed by the data with the carbohydrate inhibitors, whereas in the revision we switched this following a comment from R#1. However the sequence in which data are presented in the paper was not the sequence in which they were actually done.

The reviewer states “given there is no evidence provided for a role for Gal1 in these processes at all” but we do not know what ‘these processes’ are referring to. The hypothesis that Galectin-1 [from either source- extracellular/cell surface or intracellular] could contribute to BCP-ALL survival was based on a number of lines of evidence. One is its contribution to other cancers:  PMID 32590026, 31127849, 26348206 and many others. Our own data using a cell-permeable inhibitor specific for Galectin-1 (Paz et al 2018 PMID 29580262) demonstrated that Galectin-1 is important for BCP-ALL cells. Thus although it was reasonable to hypothesize that “inhibition of Galectin-1” could chemosensitize BCP-ALL cells, prior data did not clearly discriminate endogenous from extracellular Galectin-1.

To further emphasize the point that Galectin-1 could be important for BCP-ALL cells we have added a paragraph to the results in which literature data are cited. We also now show a new figure, S5 that illustrates that BCP-ALL cells have binding sites on the surface for extracellular Galectin-1. We also show a new Western blot for Galectin-1 in which two BCP-ALLs were treated with GM-CT-01 and GR-MD-01 (new Figure S5B).

2) The removal of the original Fig 2c (now Fig 5) means there is now no confirmation that Gal3 is knocked out in the double ko cells. And having seen this figure initially there remains the question why WT2 cells do not appear to have any Gal3 expression.

Response: We agree with the reviewer that in some types of experiments it is critical to verify knockout. However in our case these cells were derived from double knockout mice and not using shRNA, sgRNA or siRNA on wild type cells. This means that there is no possibility of the cells being a mixture, with some cells having Galectin-3 [or -1] knocked out and some not.

Regarding the lack of Galectin-3 in the WT cells, our results in general show that Galectin-3 is most highly induced as a consequence of cellular stress in BCP-ALL cells. In the absence of stress endogenous expression is low and Galectin-3 is mainly acquired from stromal cells if they are present and the cells make contact. Lack of endogenous Galectin-3 can be seen for example in Fig. 1C where the mRNA expression of Galectin-3 is about 10-fold lower than Galectin-1, in Fig. 8 and in particular in Figure S9 which makes this point. Thus it is expected that there are low levels of Galectin-3 in the WT BCP-ALL cells.

Round 3

Reviewer 3 Report

The authors have responded adequately.